# The Association Between the Perception of Exercise Benefits and Barriers and Exercise Self-Efficacy During the Induction Phase of Dialysis in Patients with End-Stage Kidney Disease: A Cross-Sectional Study

**DOI:** 10.3390/jcm13216332

**Published:** 2024-10-23

**Authors:** Yuma Hirano, Tomoyuki Fujikura, Tomoya Yamaguchi, Akihiko Kato, Kenichi Kono, Naro Ohashi, Hideo Yasuda, Katsuya Yamauchi

**Affiliations:** 1Department of Rehabilitation Medicine, Hamamatsu University Hospital, 1-20-1 Handayama, Chuo-ku, Hamamatsu, Shizuoka 431-3192, Japan; 2Internal Medicine 1, Hamamatsu University Hospital, 1-20-1 Handayama, Chuo-ku, Hamamatsu, Shizuoka 431-3192, Japan; 3Department of Physical Therapy, International University of Health and Welfare School of Health Science at Narita, 4-3, Kozunomori, Narita, Chiba 286-8686, Japan

**Keywords:** exercise self-efficacy, perceptions of exercise benefits and barriers, induction phase of dialysis, end-stage kidney disease

## Abstract

**Background/Objectives:** The physical function of patients with chronic kidney disease gradually declines as kidney function deteriorates, and this decline becomes more pronounced after the initiation of dialysis. Encouraging the development of exercise habits from the initiation phase of dialysis is crucial. Increased exercise self-efficacy is essential for establishing these habits. However, the related factors at this stage are unclear. This study hypothesized that perceptions of exercise benefits and barriers are related to exercise self-efficacy and aimed to investigate this association. **Methods:** This single-center, cross-sectional study included 72 patients and assessed the stages of exercise behavior change, perceptions of exercise benefits and barriers, and exercise self-efficacy. Multiple regression analysis was used to examine the association between exercise self-efficacy and perceptions of the benefits and barriers of exercise. **Results:** Perceptions of exercise benefits were still significantly associated with self-efficacy for exercise after adjustment for age, gender, history of cardiovascular disease, diabetic kidney disease, living alone, employment, and physical function (β = 0.474, *p* < 0.001). Similarly, perceptions of exercise barriers were also significantly associated with self-efficacy for exercise (β = −0.410, *p* = 0.001). A subgroup analysis that only examined participants without exercise habits revealed a similar association. **Conclusions:** Perceptions of exercise benefits and barriers may be associated with exercise self-efficacy in patients with end-stage kidney disease during the initiation phase of dialysis. As patients are temporarily hospitalized during this phase, it provides an opportunity for intervention. Exercise education and supportive environments during hospitalization may enhance perceptions of exercise benefits, reduce perceptions of exercise barriers, and improve exercise self-efficacy.

## 1. Introduction

In patients on dialysis, exercise habits improve the quality of life, physical function, and sleep quality and reduce the risk of death [1]. High levels of physical activity, including exercise, are also strongly associated with a lower risk of mortality [2], making it important to develop exercise habits. Nonetheless, the percentage of patients on dialysis who exercise regularly is lower (approximately 20%) compared with that of the general older population (approximately 30–40%) [3,4]. One factor contributing to this is poor physical function [5]. Physical function in patients with chronic kidney disease gradually declines as kidney function deteriorates [6], with a more pronounced decline following the initiation of dialysis [7,8]. Therefore, it is essential to encourage the establishment of exercise habits from the early stages of dialysis initiation.

To develop exercise habits, it is effective to enhance exercise self-efficacy and promote progression through individual stages of behavioral change [9,10]. Self-efficacy mediates the association between renal knowledge and self-management behaviors [11,12] and is related to exercise habits in patients on dialysis [13]. In addition to individual characteristics, such as age, gender, and physical function, cognitive aspects of goal-directed behavior are important factors related to self-efficacy, as reported in previous studies on healthy adults and chronically ill individuals [14,15,16,17]. However, the factors influencing exercise self-efficacy during the induction phase of dialysis remain unknown.

In general, understanding the benefits of exercise and perceiving fewer exercise barriers enhance the central concept of self-effectiveness, defined as “an individual’s judgment and confidence in his or her ability to successfully achieve behavioral goals or resolve difficulties” [17]. In health behavior models, one factor that influences exercise self-efficacy is the perception of exercise benefits [18], and these associations have been reported in healthy adults [19]. Although dialysis patients are aware of the benefits of exercise, they also recognize significant barriers to engaging in it [20], which contributes to low exercise adherence. However, the relationship between these barriers and exercise self-efficacy remains unclear.

We hypothesized that perceptions of exercise benefits and barriers are related to exercise self-efficacy, and this study aimed to determine this association. This study contributes to the development of exercise education and environmental setting interventions to enhance exercise self-efficacy in patients with end-stage kidney disease (ESKD) during the induction phase of dialysis.

## 2. Materials and Methods

### 2.1. Study Design, Participants, and Sample Size

This cross-sectional study included patients with end-stage renal failure who were admitted to Hamamatsu University Hospital between September 2022 and February 2024 for vascular access or hemodialysis. The inclusion criteria required participants to be 18 years of age or older. The exclusion criteria were as follows: (1) an inability to provide consent to participate in this study, (2) difficulty in assessment owing to poor general condition, (3) difficulty in assessment owing to intellectual disability or language problems, (4) missing data, and (5) cognitive decline, as shown by Mini-Mental State Examination score of ≤23. The sample size was calculated using G Power 4. Considering the linear multiple regression, effect size f^2^ of 0.26, alpha err prob of 0.05, power of 0.8, and number of predictors of 8, the required sample size was 66 [21]. All the participants provided written informed consent before participating in this study. This study was approved by the Clinical Research Ethics Committee of the Hamamatsu University School of Medicine (22-082). This cross-sectional study followed the “Strengthening the Reporting of Observational Studies in Epidemiology” guidelines.

### 2.2. Data Collection Methods and Tools

Baseline characteristics, such as age, gender, height, body weight, body mass index (BMI), primary end-stage renal disease (ESRD), comorbidities (hypertension, dyslipidemia), cardiovascular disease (CVD), smoking, alcohol consumption, employment, living alone, and the stage of exercise behavior change, were collected. The stages of exercise behavior change consisted of five items that measured actual exercise behavior in the past and present and the state of motivational readiness for that exercise behavior [22]. These items were as follows: “I am not currently exercising and do not intend to exercise in the future (precontemplation)”, “I am not currently exercising, but I plan to start in the near future (within the next 6 months) (contemplation)”, “I am currently exercising, but not regularly (preparation)”, “I am currently exercising, but have only been doing so for less than 6 months (action)”, and “I am currently exercising and have been doing so for more than 6 months (maintenance)”. The participants selected one of the items that best corresponded to their current self, and the stage of exercise transformation was determined. Precontemplation, contemplation, and preparation were not defined as exercise habits, whereas action and maintenance were defined as exercise habits. The reliability and internal validity of this assessment have been proven [22].

Our main exposure variable was the perception of exercise benefits and barriers [23]. This was evaluated using the Japanese Scale for Perceived Benefits and Barriers to Exercise. The scale consists of 10 items related to benefits (2 items for physical benefit, 2 items for psychological benefit, 2 items for social benefit, 2 items for weight management, and 2 items for self-improvement) and 10 items related to barriers (2 items for discomfort, 2 items for a lack of time, 2 items for a lack of social support, 2 items for a lack of motivation, and 2 items for a poor physical environment). The 10 items related to perceived benefits were as follows. The two items for physical benefit were “Cardiorespiratory endurance increases” and “Health improves”. The two items for psychological benefit were “Stress is alleviated, promoting relaxation” and “Enjoyment and pleasure are enhanced”. The two items for social benefit were “Social relationships are strengthened” and “Can be performed with friends”. The two items for weight management were “Helps maintain an optimal body weight” and “Improves physical appearance”. The two items for self-improvement were “Abilities are recognized by others” and “Provides a challenge to one’s potential”. On the other hand, the 10 items related to perceived barriers were as follows. The two items for discomfort were “Exercise induces fatigue” and “Exercise is boring”. The two items for a lack of time were “There is insufficient time” and “There is an excessive workload”. The two items for a lack of social support were “Family does not encourage it” and “There is no one to exercise with”. The two items for lack of motivation were “I have no energy” and “Lack of a reason to exercise”. The two items for a poor physical environment were “The weather is unfavorable” and “There are no available facilities”. The participants scored each item on a scale ranging from “not at all agree (1 point)” to “completely agree (5 points)”. Benefits and barriers were each scored in total (range, 10–50 points), with higher scores indicating higher perceptions of benefits and higher perceptions of barriers. The test–retest reliability and internal validity of the scale were acceptable [23].

Our main outcome was exercise self-efficacy [22]. It consisted of five questions (physical fatigue, mental stress, a lack of time, unusual circumstances, and inclement weather) and assessed confidence in exercising. Of these questions, unusual circumstances were not related to SE and were excluded from scoring. The question asked was as follows: “Are you confident that you would exercise regularly under the circumstances described in the following items?”. Each item was scored on a 5-point scale, ranging from “not at all (1 point)” to “very much (5 points)”, with the total score ranging from 4 to 20 points. The test–retest reliability and internal validity of this question assessment were validated [22]. Self-efficacy, as assessed using this index, was involved in exercise persistence in patients on dialysis [13].

In addition to the participant characteristics, physical function was assessed as a potential confounding factor related to the outcome and exposure variables. Physical function was assessed based on the Short Physical Performance Battery (SPPB) score [24]. This includes objective performance measures of muscle strength (five chair stands), gait speed (4 m walking speed), and balance (closed-leg standing, semi-tandem standing, and tandem standing). Each item was scored on a scale of 0–4, which, when combined, yielded a physical function score ranging from a minimum of 0 to a maximum of 12. The SPPB is commonly used to assess physical function in patients on dialysis, and a low SPPB score is associated with an increased risk of hospitalization and mortality [25]. Studies in healthy older participants have also reported an association between the SPPB score and exercise self-efficacy [16], and muscle strength, a component of the SPPB, is associated with perceptions of exercise benefits and barriers [26].

### 2.3. Statistical Analyses

In this study, participants who met the exclusion criteria were not included in the analysis. The baseline characteristics for all participants, those with exercise habits and those without exercise habits, are displayed with descriptive statistics. Single and multiple regression analyses were performed to determine the effect of the perceptions of exercise benefits and barriers on exercise self-efficacy. To control for potential confounders that may influence exercise self-efficacy, multiple regression analyses were adjusted for age, gender, a history of CVD, diabetic kidney disease (DKD), living alone, employment status, and SPPB score. These variables were measurable in this study and in previous studies have been reported to be associated with exercise self-efficacy [14,15,16,27,28,29,30]. Collinearity was confirmed using the variance inflation factor (VIF). VIFs < 10 were selected as explanatory variables, reflecting a lack of collinearity. Multiple regression analysis was also performed for a subgroup of participants without exercise habits. Non-exercising participants were selected for subgroup analysis because many patients on dialysis do not exercise, and to test the characteristics of this population. In addition, there may be differences in perceptions of exercise benefits and barriers depending on whether individuals naturally exercise [31,32]. To further characterize the perceptions of exercise benefits and barriers among the participants in this study, each item was presented using descriptive statistics. Statistical analysis was performed using IBM SPSS Statistics version 26.0, with statistical significance set at *p* < 0.05.

## 3. Results

### 3.1. Characteristics of the Survey Participants

Of the 90 patients who met the selection criteria, 81 were evaluable, excluding 7 patients with poor general condition, 1 with intellectual disability, and 1 with difficulty speaking Japanese, and 72 were included in the analysis, excluding 4 patients with missing data and 5 patients with cognitive impairment (Figure 1).

Of the participants included in this study, 69.4% were male, with a median age of 76.0 (interquartile range [IQR], 66.3–82.8) years. DKD was the most common cause of ESRD (37.5%), followed by nephrosclerosis (16.7%). The mean of exercise self-efficacy was 10.9 ± 4.4, the mean of perceptions of exercise benefits was 34.5 ± 7.4, and the mean of perceptions of exercise barriers was 22.5 ± 6.9. The median SPPB score was 12.0 (IQR, 8.0–12.0) (Table 1).

BMI is body mass index; DM is diabetes mellitus; CGN is chronic glomerulonephritis; CVD is cardiovascular disease; BUN is blood urea nitrogen; eGFR is estimated glomerular filtration rate; and SPPB is Short Physical Performance Battery.

Exercise behavior change was distributed as precontemplation (18.0%), contemplation (45.8%), preparation (16.7%), action (4.2%), and maintenance (15.3%), with many participants (80.6%) having no exercise habits (Figure 2).

### 3.2. Association Between Exercise Self-Efficacy and Perceptions of Exercise Benefits and Barriers

Single regression analysis showed that perceptions of exercise benefits were significantly associated with exercise self-efficacy (β = 0.468, *p* < 0.001) (Table 2) and that perceptions of exercise barriers were also significantly associated with exercise self-efficacy (β = −0.454, *p* < 0.001) (Table 3). Perceptions of exercise benefits were still significantly associated with exercise self-efficacy after adjusting for age, gender, history of CVD, DKD, living alone, employment, and SPPB score (β = 0.474, *p* < 0.001). Among the variables adjusted for confounding factors—such as gender, CVD, DKD, living alone, and employment status, which have been associated with exercise self-efficacy in previous studies—none showed a significant association. After adjustment for similar variables, perceptions of exercise barriers were also significantly associated with exercise self-efficacy (β = −0.410, *p* = 0.001). However, none of the adjusted confounding factors showed a significant association with exercise self-efficacy. The adjusted R^2^ values for each multiple regression model were 0.277 and 0.205, respectively (Table 2 and Table 3). In a subgroup analysis of participants without exercise habits only, multiple regression analysis showed that perceptions of exercise benefits were significantly associated with exercise self-efficacy (β = 0.486, *p* < 0.001) (Table 2). Perceptions of exercise barriers were also significantly associated with exercise self-efficacy (β = −0.396, *p* = 0.006) (Table 3).

### 3.3. Distribution of Each Item in Perceptions of Exercise Benefits and Barriers

Regarding the perception of exercise benefits, the highest scores were for physical benefits (8.5 ± 1.5), followed by psychological benefits (6.9 ± 2.1), social benefits (6.8 ± 2.5), and weight management (6.6 ± 2.0), with self-improvement receiving the lowest score (5.9 ± 2.0).

In terms of perceived exercise barriers, the highest scores were for a lack of motivation (6.0 ± 2.3), followed by a poor physical environment (4.6 ± 2.5), discomfort (4.5 ± 2.1), and a lack of social support (3.8 ± 2.0), with a lack of time receiving the lowest score (3.6 ± 2.1) (Figure 3).

## 4. Discussion

The results of this study showed that perceptions of exercise benefits and barriers were significantly associated with exercise self-efficacy, even after adjusting for age, gender, a history of CVD, DKD, living alone, employment, and physical function. To the best of our knowledge, this is the first study to investigate this association. Perceptions of exercise benefits and barriers may be ameliorated by exercise education and environmental settings and thus may be a potential therapeutic target in clinical practice in the future.

### 4.1. Association Between Exercise Self-Efficacy and Perceptions of Exercise Benefits and Barriers

Self-efficacy is defined as the belief in one’s ability to assemble and execute the set of actions necessary to achieve a targeted outcome [17]. Expectations of rewards for achieving results increase beliefs about initiating and continuing actions, even in the face of difficulties [17,33]. This means that patients who perceive more exercise benefits may have higher expectations of rewards gained from their exercise habits, leading to beliefs that they will continue to exercise, in other words, increased exercise self-efficacy. In patients with chronic diseases, knowledge and understanding of goal behaviors are involved in self-efficacy [34]. Moreover, self-efficacy mediates knowledge of and behavior toward health behaviors in healthy adults [12,35], which is consistent with the results of the present study.

The components of self-efficacy include performance achievement, vicarious experiences, verbal persuasison, and physiological and emotional arousal [17]. As exercise barriers, such as negative feelings about exercise felt in the past, a lack of social support, and a lack of a physical environment, inhibit the self-efficacy component, the perception of exercise barriers may be related to self-efficacy. Social support is associated with self-effectiveness in patients with type 2 diabetes [36]. A meta-analysis of healthy adults also reported that social support influenced adherence to health behaviors [37]. Even in patients with ESKD who are in the induction phase of dialysis in an aging population, removing exercise barriers through social support may increase exercise self-efficacy.

### 4.2. Characteristics of Perceptions of Exercise Benefits and Barriers During the Induction Phase of Dialysis

Physical benefit was the highest in the perception of exercise benefits among the participants in this study. Many patients on maintenance dialysis perceive exercise as beneficial to their health [5,38], suggesting that patients in the induction phase of dialysis may perceive similar benefits as those perceived by patients on maintenance dialysis. However, despite many patients’ positive perceptions of improved health and fitness, only approximately 20% of all patients exercised, suggesting that there may be a need to perceive exercise benefits in aspects other than physical benefits. Previous studies on patients on hemodialysis and peritoneal dialysis have reported that patients with higher levels of physical activity have higher perceived psychological benefits of exercise (e.g., improved mood) [39], and it may be important to increase the perception of psychological benefits in the study population. In contrast, a lack of motivation resulted in the highest perception of exercise barriers. Previous studies examining exercise barriers in patients on maintenance dialysis have generally reported fatigue and muscle weakness [5,20,38,40], suggesting that the primary exercise barriers may differ between the induction and maintenance phases of dialysis.

### 4.3. Proposed Interventions for Perceptions of Exercise Benefits and Barriers in the Induction Phase of Dialysis

Exercise education and environmental settings may be important in the induction phase of dialysis to increase the perception of exercise benefits and reduce the perception of exercise barriers. There is a lack of knowledge about safe exercise programs and communication and counseling with healthcare providers for patients on dialysis [5,41,42]. In addition, because social support is associated with medical compliance [37], the human and physical environments surrounding the patient influence exercise barriers. In a previous study on patients on dialysis, an exercise program that included education increased the perception of exercise benefits (not only functional but also psychological and symptom improvement) [43]. Additionally, lifestyle interventions for patients with type 2 diabetes reduce barriers [44]. In light of these findings, and considering the acknowledged gaps in exercise-related skills among nephrologists and nursing staff [45], it is expected that specific and safe exercise instructions provided by healthcare professionals with expertise in exercise will increase exercise awareness and reduce exercise barriers through social support and modifications to the physical environment [46]. Currently, the number of professionals involved in the induction phase of dialysis care is limited [47], and it may be important for many professionals, including rehabilitation professionals, to be involved in this comprehensive care effort. During the induction phase of dialysis, active exercise is difficult to perform owing to uremic symptoms. However, because patients are hospitalized for a period of time due to vascular access creation or hospitalization for dialysis induction, exercise education and environmental modifications are considered feasible.

### 4.4. Limitations

This study has some limitations. First, regarding generalizability and external validity, the participants in this study roughly reflected the general population of Japanese dialysis patients in terms of age, the proportion of males, the distribution of causes of ESRD, etc. [48]. In contrast, the participants were recruited from a single institution, which limits the diversity of the sample and may not be representative of the broader international population. In addition, data collection and measurement may have been influenced by the specific practices of this single institution, potentially affecting the generalizability and external validity of the findings. Second, this was a cross-sectional study, meaning that data were collected at a single point in time and may not capture changes over time. Finally, patients with ESKD often experience multiple symptoms, and unmeasured confounders in this study may have influenced the results. These factors prevent us from establishing definitive cause-and-effect relationships.

## 5. Conclusions

Perceptions of exercise benefits and barriers may be associated with exercise self-efficacy in patients with ESKD during the induction phase of dialysis. Based on the results of this study, it is important to conduct prospective and interventional studies in the future to establish a causal relationship. During the induction phase of dialysis, patients are temporarily hospitalized, making intervention feasible. Providing exercise education and optimizing the environment during hospitalization may increase perceptions of exercise benefits, decrease perceptions of exercise barriers, and improve exercise self-efficacy.

## Figures and Tables

**Figure 1 jcm-13-06332-f001:**
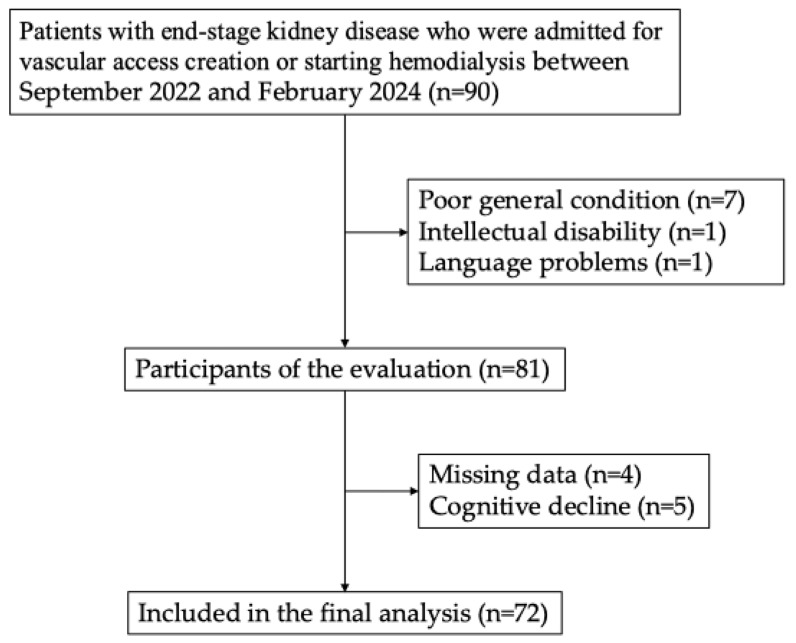
The selection of participants for analysis. Of the 90 participants included in the study period, 18 were excluded, and 72 were ultimately included in the analysis.

**Figure 2 jcm-13-06332-f002:**
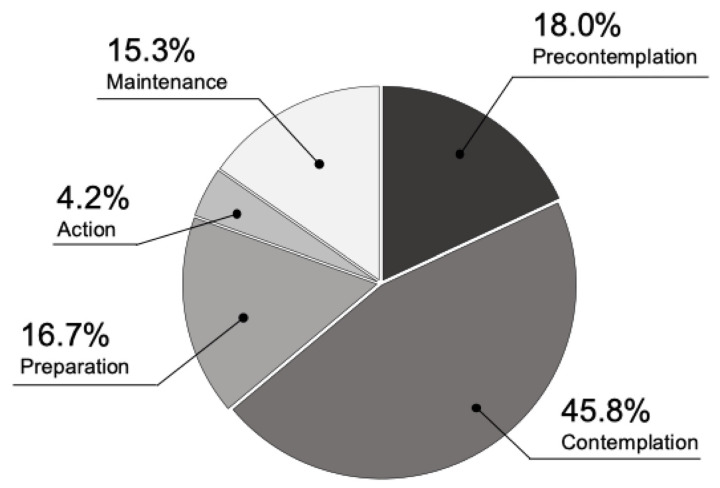
Distribution of stages of exercise behavior change in all participants. No exercise habits (precontemplation, contemplation, and preparation) accounted for approximately 80% of all respondents.

**Figure 3 jcm-13-06332-f003:**
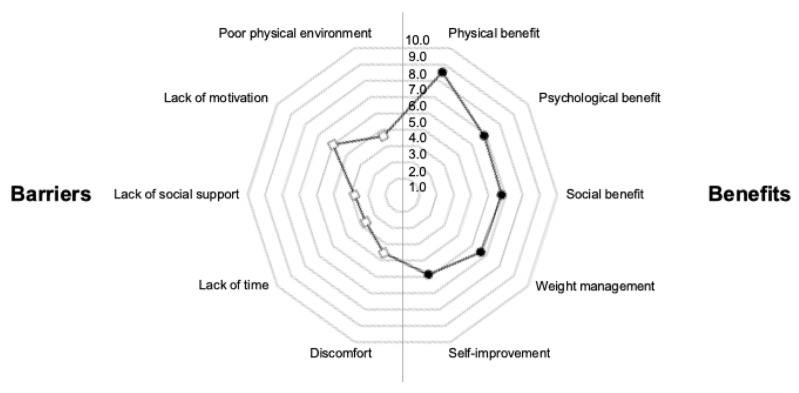
The distribution of perceptions of exercise benefits and perceptions of exercise barriers sub-items among all participants. Perceptions of exercise benefits had the highest scores for physical benefit, and perceptions of exercise barriers had the highest scores for lack of motivation.

**Table 1 jcm-13-06332-t001:** Baseline characteristics of the study participants.

	All (*n* = 72)	Exercise Habits (*n* = 14)	No Exercise Habits (*n* = 58)
Age (years)	76.0 (66.3–82.8)	78.5 (65.8–81.5)	75.0 (66.0–83.0)
Gender (*n*, [%])			
Male	50 (69.4)	13 (92.9)	37 (63.8)
Female	22 (30.6)	1 (7.1)	21 (36.2)
Height (m)	1.58 ± 0.09	1.62 ± 0.09	1.57 ± 0.09
Body weight (kg)	57.6 (50.5–67.0)	57.7 (52.6–67.1)	57.6 (49.2–67.3)
BMI	23.9 ± 4.9	22.5 ± 2.6	24.2 ± 5.2
Cause of ESRD (*n*, [%])			
DM	27 (37.5)	6 (42.9)	21 (36.2)
Nephrosclerosis	12 (16.7)	2 (14.3)	10 (17.2)
CGN	15 (20.8)	2 (14.3)	13 (22.4)
Others	18 (25.0)	4 (28.6)	14 (24.1)
Hypertension (*n*, [%])	66 (91.7)	13 (92.9)	51 (87.9)
Dyslipidemia (*n*, [%])	48 (66.7)	12 (85.7)	35 (60.3)
CVD (*n*, [%])	61 (84.7)	14 (100.0)	47 (81.0)
Current or ex-smoker (*n*, [%])	44 (61.1)	12 (85.7)	32 (55.2)
Alcohol consumption	14 (19.4)	3 (21.4)	11 (19.0)
BUN (mg/dL)	83.4 ± 23.6	81.4 ± 24.1	83.9 ± 23.4
Creatinine (mg/dL)	7.4 ± 3.4	8.0 ± 4.0	7.2 ± 3.2
eGFR (mL/min/1.73 m^2^)	6.7 ± 2.5	6.8 ± 2.8	6.6 ± 2.4
Employment	20 (27.8)	4 (28.6)	16 (27.6)
Living alone	11 (15.3)	3 (21.4)	8 (13.8)
Exercise self-efficacy	10.9 ± 4.4	13.9 ± 3.7	10.2 ± 4.3
Perceptions of exercise benefits	34.5 ± 7.4	35.8 ± 8.2	34.2 ± 7.2
Perceptions of exercise barriers	22.5 ± 6.9	19.6 ± 6.4	23.2 ± 6.9
SPPB score	10.0 ± 2.7	10.5 ± 2.0	9.9 ± 2.8

Data are expressed as means ± standard deviations, medians (interquartile ranges), and frequencies (percentages).

**Table 2 jcm-13-06332-t002:** Association between exercise self-efficacy and perceptions of exercise benefits.

	All (*n* = 72)				Subgroup (*n* = 58)			
	Univariate Regression		Multiple Regression Model *^1^		Univariate Regression		Multiple Regression Model *^2^	
	β	*p* Value	β	*p* Value	β	*p* Value	β	*p* Value
Exercise barriers	−0.454	<0.001	−0.410	0.001	−0.410	0.001	−0.396	0.006
Age			0.144	0.270			0.161	0.294
Gender			0.012	0.913			0.031	0.818
CVD			0.107	0.340			0.055	0.681
DKD			0.195	0.090			0.119	0.374
Living alone			0.013	0.904			0.040	0.766
Employment			0.191	0.131			0.173	0.252
SPPB score			0.155	0.218			0.078	0.618

CVD—cardiovascular disease; DKD—diabetic kidney disease; SPPB—Short Physical Performance Battery. *^1^ adjusted R^2^ = 0.277; *^2^ adjusted R^2^ = 0.182.

**Table 3 jcm-13-06332-t003:** Association between exercise self-efficacy and perceptions of exercise barriers.

	All (*n* = 72)				Subgroup (*n* = 58)			
	Univariate Regression		Multiple Regression Model *^1^		Univariate Regression		Multiple Regression Model *^2^	
	β	*p* Value	β	*p* Value	β	*p* Value	β	*p* Value
Exercise benefits	0.468	<0.001	0.474	<0.001	0.431	0.001	0.486	<0.001
Age			0.287	0.019			0.326	0.024
Gender			−0.111	0.318			−0.100	0.446
CVD			0.099	0.354			0.057	0.654
DKD			0.203	0.060			0.125	0.321
Living alone			0.085	0.426			0.060	0.640
Employment			0.079	0.505			0.055	0.698
SPPB score			0.265	0.029			0.259	0.090

CVD cardiovascular disease, DKD diabetic kidney disease, SPPB Short Physical Performance Battery. *^1^ adjusted R^2^ = 0.205; *^2^ adjusted R^2^ = 0.092.

## Data Availability

The data presented in this study are available on request from the corresponding author. The data are not publicly available because they are the property of the Institute of Hamamatsu University Hospital, Japan.

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
