# Peer review of "The Association Between the Perception of Exercise Benefits and Barriers and Exercise Self-Efficacy During the Induction Phase of Dialysis in Patients with End-Stage Kidney Disease: A Cross-Sectional Study"

_jcm, 2024, doi:10.3390/jcm13216332_

Round 1
Reviewer 1 Report
Comments and Suggestions for Authors
1, I have some doubts about the results of this small sample cross-sectional study
2、Although there are exclusion criteria, there is a lack of inclusion criteria
3、The participants in this study seemed to be elderly, would the title be more appropriate with the addition of OLDER?
4、Considering that the elderly have reduced physical function and multiple medical conditions, these factors may affect the results of the study, and the authors should consider this limitation
Reviewer 2 Report
Comments and Suggestions for Authors
Dear Authors,
This manuscript addresses an important topic in the management of patients with end-stage kidney disease. The study is well-designed, and the findings contribute valuable insights into the factors influencing exercise self-efficacy in this population. With minor revisions to enhance clarity and expand on certain aspects of the discussion and methods, the manuscript would be a strong candidate for publication.
1. Abstract section: Consider emphasizing the practical implications of the findings in the conclusion to better highlight the significance of the study.
2. Introduction section : The introduction could benefit from a brief mention of the broader clinical implications of improving exercise habits in this patient population, possibly linking to long-term outcomes such as quality of life or survival rates.
3. Method section:
・Clarify the rationale for the specific confounding variables included in the regression analysis. Although they are supported by previous studies, a brief justification for their inclusion in this context would strengthen the methods section.
・Consider providing more detail on how missing data were handled in the analysis.
Results section:
・ The presentation of the distribution of perceptions of exercise benefits and barriers (Figure 3) could be enhanced by including more context or interpretation directly in the results section.
・The results section could benefit from a brief discussion of any unexpected findings or non-significant results.
Discussion and Conclusion section:
・The limitations section is concise but could be expanded to include potential biases introduced by the single-center design and the cross-sectional nature of the study.
・Consider discussing the generalizability of the findings to other populations, particularly given the specific cultural and healthcare context of the study.
・Emphasize the potential for future research to build on these findings, particularly through longitudinal studies or intervention trials.
Reviewer 3 Report
Comments and Suggestions for Authors
Thank you for the opportunity to review this article. The authors aim to investigate whether there is a relationship between perceived benefits of exercise (and related barriers) and exercise self-efficacy, in people in the induction phase of dialysis.
I would like to congratulate the authors on their work. The topic is crucial because it addresses and intends to solve the problem of introducing dialysis patients to exercise programmes.
Although I appreciate the work of the authors, I feel that improvements need to be made before the article can be published.
Here are my comments:
· Abstract, line 17: the authors state that physical function declines with the start of dialysis. This assertion is erroneous, as the functional capacity of patients declines progressively as renal disease worsens. Consequently, a reduction is already present in states prior to end-stage renal disease (ESRD). I thus propose that the sentence be amended.
· Introduction, lines 47-48: please refer to the preceding comment.
· Introduction, lines 53-55: the authors cite "previous studies" but do not provide any bibliographical references. I encourage the authors to include the aforementioned studies in their work.
· Introduction, line 62: the authors assert that "patients on dialysis have a low understanding of exercise benefits". This assertion is incorrect, as evidenced by more recent studies (Ghafourifard M. et al. “Hemodialysis patients perceived exercise benefits and barriers: the association with health-related quality of life.” BMC nephrology 2021). Those undergoing dialysis are aware of the benefits of physical exercise; however, adherence to this regime is limited due to a number of identified barriers. Accordingly, it is requested that the authors reconsider the formulation of the sentence.
· Materials and Methods, line 88: the authors indicate that they collected data on the prevalence of diabetes among the participants, and do not mention CVD. Nevertheless, the data pertaining to CVD, rather than diabetes, are presented in Table 1. I would encourage them to correct the typo.
· Materials and Methods, lines 95-96: I invite the authors to remove the repetition.
· Table 1: it is recommended that the authors replace the term 'Sex' with the more appropriate term 'Gender'.
· Table 1: given that the authors have collected data on height and body weight, it would be beneficial for them to also calculate and include in the table the Body Mass Index, which is a relevant value.
· Table 1: In the section reporting the results of the SPPB, the maximum possible value of 12 is provided for all groups. I assume that this is a typo and therefore invite the authors to enter the correct values.
· Proposed interventions for perceptions of benefits and barriers in the induction phase of dialysis: the authors concentrate on the dearth of trained personnel to assist with specific exercise programmes for dialysis patients, who can provide valuable support to these individuals. In light of the acknowledged deficiencies in the specific skill sets of nephrologists and nursing staff (Regolisti G et al. “Correction: Interaction of healthcare staff's attitude with barriers to physical activity in hemodialysis patients: A quantitative assessment.” PloS one, 2018) it seems reasonable to suggest that the introduction of a dedicated professional figure into the staffing structure of the dialysis centre would be beneficial. With regard to this issue, it seems pertinent to cite a published study protocol which assesses the efficacy of an exercise expert within the context of a dialysis centre (Manfredini F et al. “A Personalized Patient-Centered Intervention to Empower through Physical Activity the Patient in the Dialysis Center: Study Protocol for a Pragmatic Nonrandomized Clinical Trial.” Methods and protocols, 2020).
Round 2
Reviewer 3 Report
Comments and Suggestions for Authors
I would like to congratulate the authors on their work. I appreciate the efforts that have been made to change the article significantly. I believe the product is now of a higher quality.
I have two further suggestions:
- Table 2: given the slight difference in BMI between the subgroups, I suggest that the authors include this parameter in the analysis of the association between exercise self-efficacy and perceived exercise benefits.
- Table 1: I suggest including the standard deviation in the SPPB data
I renew my congratulations for the work done.
